# New Perspectives in the Study of Intestinal Inflammation: Focus on the Resolution of Inflammation

**DOI:** 10.3390/ijms22052605

**Published:** 2021-03-05

**Authors:** Miguel Camba-Gómez, Oreste Gualillo, Javier Conde-Aranda

**Affiliations:** 1Molecular and Cellular Gastroenterology, Health Research Institute of Santiago de Compostela (IDIS), 15706 Santiago de Compostela, Spain; miguel.camba.gomez@sergas.es; 2SERGAS (Servizo Galego de Saúde) and IDIS (Instituto de Investigación Sanitaria de Santiago), The NEIRID Lab (Neuroendocrine Interactions in Rheumatology and Inflammatory Diseases), Research Laboratory 9, Santiago University Clinical Hospital, 15706 Santiago de Compostela, Spain; oreste.gualillo@sergas.es

**Keywords:** inflammation, macrophages, neutrophils, inflammatory bowel disease, resolution of inflammation

## Abstract

Inflammation is an essential physiological process that is directed to the protection of the organism against invading pathogens or tissue trauma. Most of the existing knowledge related to inflammation is focused on the factors and mechanisms that drive the induction phase of this process. However, since the recognition that the resolution of the inflammation is an active and tightly regulated process, increasing evidence has shown the relevance of this process for the development of chronic inflammatory diseases, such as inflammatory bowel disease. For that reason, with this review, we aimed to summarize the most recent and interesting information related to the resolution process in the context of intestinal inflammation. We discussed the advances in the understanding of the pro-resolution at intestine level, as well as the new mediators with pro-resolutive actions that could be interesting from a therapeutic point of view.

## 1. Introduction

In physiological conditions, inflammation is a self-limited process that aimed to counteract tissue damage or infection. This process starts with an induction phase, which is characterised by a strong immune response that is required for the removal of the harmful stimuli. This pro-inflammatory response must be curtailed to avoid continuous inflammation. That resolution phase is essential in restoring tissue homeostasis once the injury or the pathogen has been eliminated [1,2]. In fact, failures in the resolution of the inflammation mechanisms have been recognised as relevant players in the development of chronic inflammatory diseases, such as asthma, rheumatoid arthritis (RA), or inflammatory bowel disease (IBD) [3].

At present, most of the knowledge that is related to the regulation of the resolution process is based on lipid mediators. However, as in the induction phase, many proteins and cell types must be at play during the resolution of inflammation. Actually, it is increasingly evident that many proteins, such as Annexin A1 (ANXA1), metalloproteinases (MMPs), atypical chemokine receptors (ACKRs), or apoptotic proteins control the tissue repair and return to homeostasis after the induction of the inflammation and the removal of the danger signal [2,4].

As in other autoimmune/inflammatory diseases, IBD is also characterised by an uncontrolled inflammatory response that affects the gastrointestinal (GI) tract. Defects in the resolution of the inflammation are also suggested to be involved in the pathophysiology of this disease. Additionally, impaired resolution could be responsible for the relapses that were observed in IBD patients [5,6,7]. For that reason, the study of the mechanisms underlying the resolution process are highly relevant for the development of novel and safer therapeutic approaches.

In this review, we discussed the most recent information related to the study of the resolution phase in the context of intestinal inflammation. We summarised and highlighted the most relevant information that could be interesting for searching new treatments for IBD patients. We also included original tables and figures.

## 2. Inflammation

Inflammation is a physiological process in response to harmful stimuli, like invading pathogens or endogenous signals, such as tissue injury or dying cells [1]. This process involves a plethora of cellular and molecular events that are intended to mitigate tissue damage. In normal conditions, acute inflammatory response is a self-limited process, followed by the resolution of inflammation and subsequent functional recovery and return to tissue homeostasis. If acute inflammation is not satisfactorily resolved, it becomes chronic, which gives rise to a series of chronic inflammatory diseases, such as asthma, RA, or IBD [3].

### 2.1. Induction of the Inflammation Phase

Although the inflammatory response presents its particularities depending on the initial stimulus and the different body location, the inflammation induction phase shows general mechanisms that are directed to allow a fast and strong immune response. These universal processes involve a series of changes, such as increased vascular permeability, leukocyte recruitment, and accumulation of pro-inflammatory cytokines release (e.g., TNFα, IL-1β or IL-6), which are aimed at removing the harmful stimulus [8].

More in detail, the inflammatory response starts with the activation of the innate immune system, which initiates the inflammatory cascade through the stimulation of pattern recognition receptors (PRRs). These receptors are able to recognize either exogenous pathogen-associated molecular patterns (PAMPs) or endogenous danger signals that are generated by cellular stress or tissue injury, namely damage-associated molecular patterns (DAMPs) [9,10]. These inducers promote the secretion of a wide range of pro-inflammatory and vasoactive mediators (e.g., chemokines, cytokines, lipid mediators, vasoactive amines, etc.) by resident macrophages and mast cells of the infected or injured tissue. Subsequently, the activation of the vascular endothelium and the increase in cell adhesion molecules promotes the exudation of inflammatory proteins and the influx of leukocytes (predominantly neutrophils) to the site of inflammation [11]. Typically, neutrophils have been considered to be relevant players during the acute phase due to their efficient antimicrobial activities and tissue debris elimination capacity, mainly through these three mechanisms: phagocytosis, the production of reactive oxygen species (ROS) or proteases, and by the formation of neutrophil extracellular traps (NETs). However, recent studies revealed that the participation of neutrophils in the induction phase of the inflammation is highly intricate. It has been demonstrated that this cell type can respond to different stimuli, and consequently, neutrophils produce cytokines, which are able to modulate inflammation and the activity of other immune cells [12,13]. Ideally, once neutrophils complete their functions undergo apoptosis, followed by their elimination by macrophages favouring the resolution of inflammation. If this process fails and the inflammatory response persists, the immune cells infiltrate change, which leads to the incorporation of macrophages and T cells. Although the composition of the infiltrate could vary, depending on the trigger (e.g., pathogens, autoimmune responses, the presence of self-antigens, etc.). In any case, some of the effectors that are released by these cells, such as ROS, are unable to differentiate between pathogen and host tissues, a situation that generates undesired collateral damage [14]. For this reason, the control of inflammatory response is essential in avoiding the development of chronic inflammatory diseases or cancer [15].

### 2.2. Resolution of the Inflammation Phase

Once the harmful stimuli are eliminated, the inflammatory process enters in the resolution phase. This mechanism is directed to curtail inflammation, leading to the repair and functional recovery of the tissue. This phase is vital to the control of the excessive pro-inflammatory response and the restoration of tissue homeostasis. Classically, the resolution process was considered to be a passive mechanism. However, specific molecular and cellular events that regulate this process have recently been described, demonstrating that the resolution of inflammation is an active and highly regulated process. In fact, there are well defined key mechanisms, which are common across different tissues and diseases [2,16]. These three general aspects of resolution are [2,16]:(1)Cessation of neutrophil influx: neutrophils are the most abundant leukocytes at the site of inflammation. The initial step for the resolution of inflammation depends on stopping the infiltration of this cell type.(2)Neutrophil apoptosis and removal: different signals induce neutrophil apoptosis, which, in turn, are removed, mainly by macrophages in a process termed efferocytosis.(3)Macrophage shift: during resolution of inflammation, this cell type changes its pro-inflammatory phenotype and acquires anti- inflammatory and pro-resolving functions.

Like in the induction phase, these different steps that are involved in the resolution of inflammation are controlled by a wide variety of mediators, including exclusive lipid and protein pro-resolving mediators [e.g., lipoxins, specialized pro-resolving lipid mediators (SPMs) or ANXA1] [2,4]. However, many proteinases, chemokines, receptors, and cytokines, which are not exclusively involved in the resolution process, contribute to the down-regulation of pro-inflammatory signals and return to tissue homeostasis [4,17,18].

As mentioned above, the cessation of neutrophil extravasation to the site of inflammation is critical for the induction of resolution. This process is controlled at various levels and it involves the participation of lipid and protein effectors. Regarding lipid mediators, it has been demonstrated that SPMs play pivotal roles in neutrophil recruitment inhibition. For instance, the SPMs Maresin 1 (MaR1) and Maresin 2 (MaR2) are able to decrease neutrophil infiltration in mouse zymosan-induced peritonitis [19]. Additionally, protectin D1n-3 docosapentaenoic acid (DPA) and resolving D5n-3 DPA exert similar actions during peritonitis and intestinal inflammation, and this effect is due to the regulation of neutrophil adhesion to the endothelium [20]. In line with this, one of the major protein components of resolution cascade, ANXA1, has been extensively studied by its actions on neutrophil infiltration blockade in acute models of inflammation, such as acute murine MSU-induced gout [21], DSS-induced colitis [22], zymosan-induced peritonitis [23], among others [24]. These studies reveal a prominent role for ANXA1 on neutrophil infiltration, by the inhibition of neutrophil adhesion to the endothelium [25] and the impairment of adequate rolling [26] and extravasation to the sites of inflammation [27].

Chemokine and cytokine gradient reduction is also necessary for the regulation of neutrophil influx. In this scenario, MMPs have shown to control the activity and the availability of these relevant inflammatory mediators and, consequently, regulate neutrophil recruitment. Apart from their well-known actions on the regulation of the extracellular matrix (ECM) remodelling, MMPs proteolytic activity is also able to control the function of potent pro-inflammatory cytokines, such as TNF-α or IL-1β [28,29,30]. Moreover, the proteolysis of chemokines or their accessory binding molecules on the ECM causes the inactivation of these chemoattractant proteins, which impairs neutrophil migration [31,32]. In line with this, chemokine activity is also modulated by its binding to ACKRs. These are a group of non-conventional chemokine receptors that, instead of promoting migratory signals, induce a change in chemokine gradient by the sequestration of these proteins from the extracellular compartment [18,33]. For that reason, ACKRs have emerged as relevant players in the generation of a pro-resolutive environment.

The death and removal of the exhausted neutrophils is another prominent pro-resolutive signal. Like many other cells, apoptosis can be induced in neutrophils by binding to apoptosis ligands, such as Fas ligand (FasL) and TNF-related apoptosis-inducing ligand (TRAIL), which are produced by different cell types, such as macrophages or neutrophils [34,35,36]. Apoptotic cells can then be engulfed by phagocytes in a process called efferocytosis. This process is facilitated by the presence of eat-me signals in the membrane of apoptotic neutrophils. One of the most common eat-me signals is the expression of phosphatidylserine, which is recognised by different receptors, such as T cell immunoglobulin and mucin domain containing 4 (TIM4) or brain-specific angiogenesis inhibitor 1 (BAI1) [37,38,39]. Efferocytosis induces the cessation of the pro-inflammatory stimulus [40], but it also stimulates immunosuppressive effects in the phagocytes by the production of anti-inflammatory and pro-resolving mediators, such as IL-10 and SPMs [41,42,43]. This switch in macrophage phenotype is needed for a proper resolution of inflammation. Indeed, these cells acquire essential anti-inflammatory properties, which promote resolution [5] (Figure 1).

## 3. Inflammatory Bowel Disease

Inflammatory bowel disease (IBD), which is mainly comprised of Crohn’s disease (CD) and ulcerative colitis (UC), is a group of immune-mediated disorders of the GI tract, characterized by chronic and recurring inflammatory response in the digestive tract mucosa. While UC is restricted to the colon and presents a continuous inflammation confined to the mucosal layer, CD is characterized by a discontinuous and transmural inflammation that can occur anywhere in the GI tract. This disease cannot be cured and, despite the current therapies, a proportion of patients suffers from relapses and continuous inflammation, which, in some cases, lead to the surgical removal of parts of the intestine. The peak onset of the disease, which ranges from 15–35 years, affects patients in a highly productive age. Typical IBD symptoms, including such as pain, diarrhoea, fever, etc., interfere with patients’ daily tasks. Moreover, suffering from a lifelong disease can be an emotional burden. For that reason, some IBD patients present anxiety, anger, or fear. All of these factors, together with the high costs for the Health System, make this disease a huge socio-economic problem.

IBD incidence increased in the second half of the 20th century and the beginning of the 21st century, especially in developed countries. Although, over the last decades, the incidence of this disease was shown to be stable in western countries. An increased incidence rate in newly industrialized countries of Asia, Africa, or South America was observed [44].

The primary cause that initiates IBD still remains unclear. However, it is generally accepted that an uncontrolled inflammatory response, most likely driven by the microbiome and defective barrier functions, promotes a vicious cycle that leads to chronic disease. Additionally, genetic and environmental factors are believed to contribute to the pathogenesis of IBD.

The adaptive immune system has been typically considered to be the most relevant player in the development of intestinal inflammation. However, at present, it is generally accepted that the innate immune system is equally important for the regulation and maintenance of gut homeostasis. Macrophages and neutrophils (essential components of the innate immune system) have both been reported to be involved in the control of the inflammatory response at the intestinal level in IBD animal models and patients [45,46].

## 4. Factors Involved in the Resolution of Inflammation in IBD

Recent evidence indicates that defects in the mechanisms that are involved in the resolution of inflammation can result in the perpetuation of a persistent and uncontrolled pro-inflammatory response in the intestinal tissue and, as a consequence, this impaired ability to resolve inflammation could contribute to the onset and progression of IBD [5,6,7]. Hereafter, we will discuss the recent advances of different pro-resolutive factors and mechanisms in the context of intestinal inflammation.

### 4.1. Specialized Pro-Resolving Mediators

SPMs are a large class of signalling molecules derived from the arachidonic acid (AA), called lipoxins (LXs), or from the omega-3 polyunsaturated fatty acids (omega-3-PUFA), named resolvins (Rvs), protectins (PDs), and maresins (MaRs) [47,48,49]. These molecules can be produced by the cells of the innate system and non-immune cells and, in general, they decrease leukocyte reactivity and promote tissue repair and regeneration [50,51] (Table 1).

Lipoxins is the first recognized family of SPMs and one of the most studied. The role of lipoxin A4 (LXA4) has been extensively studied in the context of intestinal inflammation. The severity of dextran sodium sulphate (DSS)-induced colitis and trinitrobenzene sulfonic acid (TNBS)-induced colitis, as well as the expression of several pro-inflammatory cytokines at the colonic level, were demonstrated to be reduced by the administration of LXA4 analogues [52,53]. More recently, similar effects were described after the administration of another LXA4 analogue in mice presenting Crohn´s-like intestinal lesions that are produced by the administration of cholate-containing high fat diet and the knockout of cyclooxygenase 2 (Cox2) in myeloid cells [54]. In addition, the knockout mice for formyl peptide receptor 2/3 (fpr2/3), orthologs of the human LXA4 receptor FRP2, showed a delayed mucosal healing and increased mucosal ulceration in comparison to their wild type counterparts during the course of acute DSS-induced colitis [55]. Taken together, these results highlight the potential protective effects of LXA4 in the development of intestinal inflammation. In fact, the LXA4 levels were elevated in colon biopsies from UC patients in remission [56], which suggests a role for this pro-resolutive mediator in colon tissue recovery and homeostasis.

Maresins (macrophage mediators in resolving inflammation) are the SPMs first identified in macrophages [57], but also produced by neutrophils [58], which are biosynthesised from docosahexaenoic acid (DHA) and by the action of 12-lipoxigenase (12-LOX) into MaR1 and MaR2 [19,57,59]. MaR1 is the best-known maresin, and its actions on the resolution of inflammation are well documented in different tissues and diseases [16,60]. Regarding IBD, it has been demonstrated that MaR1 exerted anti-inflammatory and pro-resolving actions in two different animal models of colitis (DSS- and TNBS-induced colitis) [61]. Mice that were treated with MaR1 showed decreased disease severity, which was accompanied by a reduction in the infiltration of polymorphonuclear leukocytes (PMNs) into the colon and by the down-regulation in the expression of different pro-inflammatory cytokines [61,62]. These MaR1 effects on colitis development were thought to be mediated by the activation of nuclear factor erythroid 2-related factor 2 (Nrf2) signalling, the inhibition of nuclear factor (NF)-κB signalling pathway, and the enhancement of the differentiation of M2 anti-inflammatory macrophages [61,62].

Resolvins are also lipid mediators that are derived from the eicosapentaenoic acid (EPA) or DHA, resulting in E-series resolvins (RvE) and D-series resolvins (RvD), respectively. Like maresins, different members of this family were demonstrated to be involved in the regulation of colitis development in vivo. RvE1 treatment caused a decrease in the severity of the intestinal inflammation in mice that were subjected to both DSS and TNBS-induced colitis [63,64,65]. This milder colitis was accompanied by a decrease in the infiltration of PMNs and the down-regulation of several pro-inflammatory mediators in the colon [63,65]. In the same way, RvD2 or RvD5n-3 DPA treated mice presented reduced body weight loss, colon shortening, and histologic severity scores when compared to vehicle-treated mice during acute DSS-induced colitis [20,66]. Additionally, as observed for RvE1, the administration of RvD2 restrained the production of pro-inflammatory cytokines, such as TNF-α or IL-1β, and decreased the phosphorylation of the NF-κB subunit p65 in colon tissue [63,66], suggesting that these SPMs could dampen inflammation through the modulation of the canonical NF-κB signalling pathway. More recently, Quiros M et al., published an interesting study, in which RvE1 promoted intestinal mucosa wound repair by increasing epithelial cell migration and proliferation [67]. Altogether, these findings show that resolvins are not only able to control inflammation by decreasing excessive inflammatory cell infiltration and pro-inflammatory factors production. Additionally, these SPMs activate pathways, leading to a correct repair of the mucosal surfaces, which is essential for the maintaining of colon homeostasis and protection against pathogens.

Apart from D-series resolvins, protectins are also derived from the metabolisation of DHA by the 15-LOX enzyme [68]. In line with the other families of SPMs, protectins were found to act as anti-inflammatory and pro-resolving mediators in different animal models of inflammation [69,70,71]. In the last years, several studies have also demonstrated the participation of these SPMs in the regulation of intestinal inflammation. PD1 treatment to mice that were subjected to DSS-induced colitis, together with an eosinophil depletion, prevented colon shortening and neutrophil infiltration, improved colonic histology scores, and reduced the expression of pro-inflammatory and chemotactic factors, such as TNF-α, IL-1β, CXCL1, and CXCL2 [71]. Similarly, PD1 exogenous administration reduced the severity of antibiotic-induced colon inflammation [72]. PD1 treatment also caused a prevention of shortening of the colon length, as well as an amelioration in the epithelial damage and histologic inflammation scores, as observed in the DSS colitis model [72]. Finally, a congenerous of protectins family produced by the metabolism of n-3 docosapentaenoic acid (n-3 DPA), namely PD1n-3 DPA, was reported to protect mice from inflammation during the course of acute DSS-induced colitis [20]. Additionally, in the same study, the authors demonstrated that the mechanism underlying that observed protection against colitis development involved the regulation of leukocyte trafficking. Particularly, PD1n-3 DPA seemed to inhibit the adherence of neutrophils to the endothelium [20]. Taken together, all of these data reveal a prominent role for protectins in the control of exacerbated intestinal inflammation by the inhibition of neutrophil recruitment into the inflamed colon tissue.

### 4.2. Annexin A1

Annexin A1 (ANXA1) is a 37-kDa phospholipid-binding protein that acts as a downstream effector of glucocorticoids (GCs). This protein is a well-characterized anti-inflammatory and pro-resolving factor [73,74], which is able to blockade leukocyte recruitment to inflamed tissues [23], inhibit pro-inflammatory mediators release [75], and promote tissue repair [76].

The participation of ANXA1 in IBD has been extensively studied and many works highlight the relevance of this pro-resolutive factor in the development and treatment of intestinal inflammation (Table 2). Pre-clinical studies have demonstrated that AnxA1 KO mice presented increased susceptibility to DSS-induced colitis, showing worse histologic scores and increased mucosal injury than the respective wild type controls [22]. Moreover, the colonic epithelium repair after DSS withdrawal was impaired in AnxA1-deficient mice [22], which suggested that this protein is involved in both the protection against intestinal inflammation and the promotion of repair processes. Several years after the publication of that study, a few studies corroborated the therapeutic potential of ANXA1. The administration of MC-12, an ANXA1 based peptide, was shown to be effective in the amelioration of colitis symptoms in DSS and TNBS IBD models [77]. Mice that were treated with MC-12 presented less colonic damage, reduced levels of myeloperoxidase (MPO), and decreased expression of TNF-α, IL-1β, IFN-γ, IL-6, and PGE2. Moreover, these changes were accompanied by a strong decrease in the activation of NF-κB [77], and, given the relevance of this signalling pathway in the control of the inflammatory response at intestinal level, that observed inhibition could explain the anti-inflammatory effects of MC-12 in both IBD mice models. Additionally, the therapeutic administration of nanoparticles containing the AnxA1 mimetic peptide Ac2-26 after DSS withdrawal produced a faster recovery of the colitis symptoms as compared to mice that were inoculated with nanoparticles containing a scrambled peptide [78]. Accordingly, intramucosal treatment with the Ac2-26 nanoparticles showed accelerated intestinal wound healing [78], demonstrating the prominent role that is played by ANXA1 in the recovery of colonic epithelial injury and return to tissue homeostasis. Li C et al. further corroborated these therapeutic effects of ANXA1, who extensively studied the actions of oxidation-responsive nanoparticles containing Ac2-26 (namely AON by the authors) in the development of intestinal inflammation [79]. The inoculation of those nanoparticles allowed for a site-specific accumulation of Ac2-26 in the inflamed colons, showing a great efficacy in the amelioration of IBD symptoms in vivo. More in detail, the administration of AON showed therapeutic effects to mice that were subjected to both acute and chronic DSS colitis and IL10-deficient mice, which spontaneously develop intestinal inflammation [79]. In all of the IBD models tested, AON-treated mice presented low body weight loss, improved disease activity index, and reduced mucosal damage. Additionally, MPO activity and the expression of TNF-α, IL-1β, and IFN-γ were down-regulated at the colon level after the inoculation of AON. The authors also explored the mechanisms behind AON therapy, and they observed that key aspects of the resolution of inflammation process were regulated by the nanoparticles containing this ANXA1 mimetic peptide. Among these, AON was found to inhibit the production of ROS and TNF-α in macrophages, as well as induce a phenotype switching to M2-like macrophages. Moreover, neutrophils displayed a decreased migration capacity and increased efferocytosis by macrophages when treated with AON [79].

The therapeutic potential of ANXA1 was also suggested in the clinical setting. ANXA1 mRNA levels in peripheral blood mononuclear cells (PBMCs) were decreased in CD patients when compared to healthy individuals. Noteworthy, the expression of this resolutive factor was upregulated in patients that were treated with infliximab and clinical remission, but not in the non-responders group [79]. Moreover, ANXA1 expression in colon biopsies was increased in CU patients in clinical remission [56] and CD patients with higher intrinsic expression of this protein presented lower disease severity [80,81], which suggested the relevance of this pro-resolutive mechanism in the control of human intestinal mucosa homeostasis.

### 4.3. Chemokines Gradient Depletion by MMPs and ACKRs

Chemokines are low molecular proteins that control the recruitment of immune cells to the site of inflammation. For that, chemokine gradient depletion is considered to be critical in the cessation of neutrophil influx and the consequent promotion of the resolution process at the inflamed tissue. In this review, we wanted to focus our attention in two relevant mechanisms by which chemokine inactivation could be achieved: chemokine cleavage by MMPs and chemokine sequestration by ACKRs.

MMPs proteolytic activity is now recognized to be involved in many different physiological and pathological process, apart from the ECM remodelling [17]. The involvement of MMPs in the regulation of inflammation by the processing of chemokines has been postulated 20 years ago [82,83,84]. Since then, many studies demonstrated a major role for MMPs in the control of chemokine activity [31,85], which directly affects the infiltration and migration of leukocytes [31].

The expression and activity of different MMPs have been found altered in IBD animal models and patients [86,87,88]. Moreover, the administration of MMPs inhibitors or specific MMPs ablation were able to modulate the severity of the intestinal inflammation [89,90,91]. Altogether, these data revealed that MMPs play a role in the pathophysiology of IBD (Table 3). However, the specific effect of each MMP differ among the different members of this family of proteins. For instance, Mmp-9-deficient mice showed an amelioration of the severity of experimental colitis [90], while opposite findings were obtained in mice that were lacking Mmp-2, Mmp-10, and Mmp-19 [91,92,93]. In these three studies, mice with the ablated MMPs presented an exacerbation of colitis symptoms, which was accompanied by an increase in leukocyte infiltration in the colon. Moreover, the study of the recovery phase after DSS withdrawal demonstrated that these mice were unable to adequately resolve colon inflammation [91,92,93], highlighting that mice lacking the expression of Mmp-2, Mmp-10 and Mmp-19 have profound defects in the mechanisms of resolution of inflammation. Actually, these mice presented a dysregulated leukocyte infiltration during the recovery phase, which, in the case of Mmp-19-deficient mice, was partly linked to defects in the cleavage of the chemokine Cx3cl1 by this metalloproteinase [93].

ACKRs also participate in the control of chemokine availability and, thereby, contribute to the generation of a resolutive environment. Generally, these receptors bind chemokines with a high affinity, but do not induce migration due to their inability to engage G proteins [18,33]. Consequently, ACKRs sequestrate chemokines from the extracellular compartment and induce chemokine degradation. Although these atypical receptors are also able to form heterodimers with chemokine canonical receptors and modify their signalling [94,95].

The contribution of ACKRs to the development of IBD has been postulated (Table 3). However, there is a limited number of published articles that are related to this topic, which could be a reflection of the relatively short life of this area of knowledge. Anyway, interesting information was obtained from DSS experimental colitis and the genetic modification of different members of ACKR family. The expression of Ackr2 was found to be up-regulated in inflamed colons after DSS treatment and, surprisingly, mice lacking the expression of this receptor were less susceptible to colitis, showing less clinical symptoms and a reduced severity of intestinal inflammation histologically [96]. The observed leukocyte infiltration in wild type and Ackr2-deficient mice was similar, as well as the concentrations of multiple chemokines [96], suggesting that, in this IBD model, Ackr2 does not seem to modulate chemokine bioavailability. On the contrary, the lack of this atypical chemokine receptor confers protection against colitis through the regulation of the pro-inflammatory cytokine Il-17 [96]. More recently, very interesting findings were obtained by Song ZY et al. These authors found the colocalization of the atypical chemokine receptor ACKR3 and the conventional receptor CXCR4 in colon biopsies from colorectal cancer patients [97], which suggested the heterodimerization of both receptors. The Villin-Cxcr7-Cxcr4, Villin-Cxcr7, and Villin-Cxcr4 transgenic mice were used in order to investigate the impact of that heterodimer in colon homeostasis. Interestingly, transgenic mice for both receptors showed exacerbated intestinal inflammation when compared to both single transgenic mice [97], revealing the importance of atypical and conventional chemokine receptors heterodimers in the regulation of intestinal malignancies.

### 4.4. Neutrophil Death

Once the induction phase of the inflammation ends, the apoptosis of the neutrophils accumulated in the sites of inflammation is crucial in the restoration of tissue homeostasis. Dysregulated neutrophil apoptosis is usually linked to disease and, in the particular case of inflammatory/autoimmune diseases, a decreased neutrophil apoptosis rate is often observed [36]. In fact, increased neutrophil survival and delayed spontaneous apoptosis has been reported in IBD patients [98,99]. Similarly, ex vivo experiments using neutrophils isolated from an equine model of colitis showed delayed LPS-induced apoptosis [100]. These studies raise the idea of an increased neutrophil lifespan in the context of intestinal inflammation, which could affect the proper resolution of inflammation. Actually, the lack of the apoptotic signal Trail in mice subjected to DSS-induced colitis aggravated the severity of the disease and produced a strong infiltration of leukocytes [101]. In line with this, mice that were deficient in the apoptosis agonist BH3 interacting domain death agonist (Bid) presented more sustained weight loss than their wild type counterparts during the course of DSS acute colitis [102] (Table 4).

### 4.5. Macrophage Pro-Resolutive Phenotype

Macrophages are considered to be key elements in the maintenance of gut homeostasis. This cell type is responsible for the tolerance against commensal bacteria and food antigens and it protects the intestinal tissue against an excessive inflammatory response in response to those factors [103,104]. Moreover, macrophages are the main components of the inflammatory response and one of the most relevant players in the resolution process. In fact, once the harmful stimuli are removed, macrophage phagocytosis of dying cells and the production of cytokines and lipid mediators are essential for the restoration of the normal function of the tissue [103,104] (Table 4).

Neutrophil efferocytosis by macrophages induces an anti-inflammatory program in these cells, which results in the production of different immunoregulatory cytokines, such as TGF-β and IL-10 [105]. Both of the cytokines develop potent immunosuppressive actions in the gut. Actually, TGF-β-deficient mice show multisystemic inflammations involving the intestine [106]. Moreover, transgenic mice with impaired TGF-β signalling present a severe inflammation in the colon [107]. The anti-inflammatory effects of this cytokine in IBD were further demonstrated by using mice animal models, in which TGF-β signalling blockade exacerbated colitis symptoms [108]. On the contrary, the potentiation of its activity attenuated intestinal inflammation [109]. Recently, different studies revealed the impact of macrophage TGF-β signalling on IBD. It has been demonstrated that extracellular vesicles that are released by apoptotic cells induce the production of TGF-β by macrophages in vitro and in vivo [110]. In addition, the administration of the aforementioned extracellular vesicles to mice under T cell transfer experimental colitis improved colitis severity by a mechanism involving TGF-β, since the use of neutralizing antibodies against this cytokine abolished the therapeutic effects of the extracellular vesicles [110]. In line with this, mesenchymal stem cells (MSCs) transplantation to mice that were subjected to DSS-induced colitis reduced intestinal inflammation severity, and this effect was suggested to occur through the recruitment of macrophages with high TGF-β expression levels [111].

IL-10 is widely recognized for its role in the control of intestinal inflammation and macrophage function. IL-10 deficient mice are a well-established IBD animal model due to the development of spontaneous colitis. More importantly, the specific loss of expression of IL-10 receptor (IL-10R) in macrophages causes severe intestinal inflammation per se [112], highlighting the relevance of IL-10 signalling in macrophages for the maintenance of gut homeostasis. In fact, macrophages from mice with a specific deletion of IL-10Ra showed a marked pro-inflammatory signature, and the axis Il23/Il22 was responsible for the detected spontaneous colitis in those mice [113]. In line with the observed actions of the TGF-β, it has been reported that the macrophage-derived IL-10 was also essential for the anti-inflammatory effects of MSCs therapy during DSS-induced colitis [114]. Moreover, the disruption of the IL-10-STAT3 signalling by the phosphatase Shp2 in macrophages was responsible for the exacerbation of intestinal inflammation [115].

The engulfment of apoptotic neutrophils triggers a phenotypic change from pro-inflammatory to anti-inflammatory macrophages, as mentioned above. The beneficial effects of anti-inflammatory or M2 macrophages on colitis development have been well demonstrated. For instance, it has been shown that transfer, by intraperitoneal injections, of M2 macrophages into mice that were subjected to dinitrobenzene sulfonic acid (DNBS)-induced colitis, caused an attenuation of the disease activity and the histological damage [116]. More recently, exosomes that were derived from M2-like macrophages containing the miR-590-3p were shown to protect mice from DSS colitis [117]. Interestingly, apart from the decreased mucosal damage, these exosomes promoted wound healing and colon repair [117], highlighting the contribution of these cells to the epithelial regeneration after colon inflammation.

The metabolic regulation of macrophage function and phenotype is another aspect that has increased in relevance in the last years. Recently published studies have pointed out the importance of macrophage metabolic status and how this could revolutionize the study of resolution of inflammation. At present, it is recognized that, under the induction phase of inflammation, macrophages shift from oxidative phosphorylation to aerobic glycolysis (Warburg effect). This metabolic switch is observed in pro-inflammatory macrophages, where energy demand is highly increased to carry out inflammation. On the contrary, anti-inflammatory macrophages use the Krebs cycle for energy production [118].

Some evidence demonstrated that macrophage metabolic reprogramming could be also at play in the regulation of intestinal inflammation. A few years ago, it was reported that the IL-10 regulation of the inflammasome on macrophages were metabolic-dependent [119]. IL-10 control of macrophage function under a pro-inflammatory stimulation was related to an altered metabolic profile, in which this cytokine inhibited the glycolytic influx and prevented the accumulation of dysfunctional mitochondria via a mammalian target of rapamycin (mTOR). In that scenario, IL-10 negatively regulated inflammasome activation in bone marrow-derived macrophages, but similar observations were also obtained in lamina propria macrophages from IL10-deficient colitic mice, which suggested that mitochondrial impairment and the subsequent inflammasome activation could affect the development of intestinal inflammation [119].

Finally, another interesting study also revealed the crucial role that is played by the metabolic reprogramming on macrophage polarization. Kang, S. et al., demonstrated how defects in the axis mTOR-Semaphorin 6D (Sema 6D)-Peroxisome proliferator receptor γ (PPARγ) led to an aberrant fatty acid uptake and metabolism, which ultimately impaired the anti-inflammatory macrophage polarization [120]. In line with this, Sema 6D-deficient mice showed exacerbated DSS-induced colitis, which was characterized by a more severe leukocyte infiltration, decreased expression of IL-10 by lamina propria cells, and increased production of pro-inflammatory cytokines in the same cells [120]. Altogether, these data further support the relevance of macrophage metabolic reprogramming in the physiopathology of IBD.

## 5. Conclusions

IBD current anti-inflammatory and immunosuppressive therapies target effector pathways that are involved in the induction of inflammation. Despite the great therapeutic success of the aforementioned treatments, these drugs are often related to the risk of blunting physiological immune responses, leading to undesirable side effects, such as opportunistic infections. Moreover, a proportion of the IBD patients do not respond to certain treatments (e.g., biologic therapies) or lose effectiveness over time. Thus, there is room for the improvement of these treatments and their ability to achieve a complete resolution of the inflammation and mucosal healing.

For those reasons, there is an imperative need for the development of new and safer drugs that could reduce inflammation and promote intestinal tissue healing, avoiding relapses. In this scenario, pro-resolving therapies arise as an appealing approach, due to their ability to induce tissue repair without, in principle, increasing the risk of infection. Although the available data are promising, more information about the role played by specific pro-resolving factors and their downstream molecular and cellular mechanisms is necessary for the generation of new treatments.

## Figures and Tables

**Figure 1 ijms-22-02605-f001:**
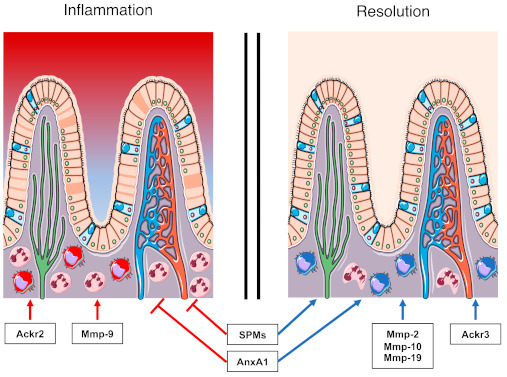
Summary of the effects of the indicated factors on the inflammatory response.

**Table 1 ijms-22-02605-t001:** Specialized pro-resolving mediators (SPMs) actions on intestinal inflammation.

Molecules	Functions	References
Lipoxin A_4_	Attenuates the severity of DSS- and TNBS-induced colitis Induces tissue repair and homeostasis Reduces the expression of pro-inflammatory cytokines	[52,53,54]
Maresin 1	Reduces DSS- and TNBS-induced colitis severity Reduces PMN infiltration Reduces the expression of pro-inflammatory cytokines Activates Nrf2 signaling pathway Inhibits NF-κB signalling pathway Enhances the differentiation of M2 anti-inflammatory macrophages	[61,62]
Resolvin E1	Reduces DSS- and TNBS-induced colitis severity Reduces PMN infiltration Reduces the expression of pro-inflammatory cytokines Improves intestinal mucosa wound repair Inhibits NF-κB signalling pathway	[63,64,65,67]
Resolvin D2	Reduces DSS-induced colitis severity Reduces the expression of pro-inflammatory cytokines Inhibits NF-κB signalling pathway	[66]
Resolvin D5_n-3_	Reduces DSS-induced colitis severity Inhibits the adherence of neutrophils to the endothelium	[20]
Protectin D1	Reduces DSS-induced colitis severity Reduces PMN infiltration Reduces the expression of pro-inflammatory cytokines	[69,70,72]
Protectin D1_n-3 DPA_	Reduces DSS-induced colitis severity Inhibits the adherence of neutrophils to the endothelium	[20]

**Table 2 ijms-22-02605-t002:** Annexin A1 actions on intestinal inflammation.

Molecules	Functions	References
Annexin A1	Attenuates the severity of DSS-induced colitis Blocks leukocyte recruitment to inflamed tissues Inhibits the release of pro-inflammatory mediators Promotes tissue repair	[22]
MC-12	Reduces DSS- and TNBS-induced colitis severity Reduces the levels of MPO Decreases the production of TNF-α, IL-1β, IFN-γ, IL-6 and PGE2 Reduces the activation of NF-κB signalling pathway	[77]
Ac2-26	Promotes tissue repair Accelerates intestinal wound healing Reduces body weight loss Reduces the severity of acute and chronic DSS-induced colitis and IL-10-deficient mice colitis Reduces MPO activity Decreases the expression of TNF-α, IL-1β and IFN-γ Inhibits the production of ROS and TNF-α in macrophages Enhances the differentiation of M2 anti-inflammatory macrophages Reduces migration capacity of neutrophils Increases efferocytosis by macrophages	[78,79]

**Table 3 ijms-22-02605-t003:** Metalloproteinases (MMPs) and atypical chemokine receptors (ACKRs) actions on intestinal inflammation.

Molecules	Functions	References
Mmp-2-defiecient mice	Aggravates DSS-induced colitis severity Reduces tissue repair Aggravates the severity of *Salmonella*-induced colitis Increases leukocyte infiltration	[91]
Mmp-9-deficient mice	Reduces DSS-induced colitis severity Reduces the severity of *Salmonella*-induced colitis	[90]
Mmp-10-deficient mice	Aggravates DSS-induced colitis severity Reduces tissue repair Increases leukocyte infiltration	[92]
Mmp-19-deficient mice	Aggravates DSS-induced colitis severity Reduces tissue repair Increases leukocyte infiltration Defects in the cleavage of the chemokine Cx3cl1	[93]
Ackr2-deficient mice	Reduces DSS-induced colitis severity Protection against colitis through the regulation of the pro-inflammatory cytokine Il-17	[96]
Ackr3-deficient mice	Villin-Cxcr7-Cxcr4 transgenic mice shows exacerbated DSS-induced colitis	[97]

**Table 4 ijms-22-02605-t004:** Neutrophil and macrophage actions on intestinal inflammation.

Molecules	Functions	References
Trail-deficient mice Bid-deficient mice	Aggravates DSS-induced colitis severity Aggravates DSS-induced weight loss	[101,102]
TGF-β	Reduces DSS- and T cell transfer-induced colitis severity Impaired TGF-β signalling produces a severe inflammation in the colon	[106,107,108,109]
IL-10	IL-10-deficient mice develop spontaneous colitis Loss of expression of IL-10R in macrophages causes severe intestinal inflammation	[112,113]
M2 macrophages	M2 macrophages transfer into mice subjected to DNBS-induced colitis attenuates colitis severity Exosomes derived from M2-like macrophages protect mice from DSS colitis, promote wound healing and colon repair	[117]
Metabolic reprograming	Pro-inflammatory macrophages use aerobic glycolysis and anti-inflammatory macrophages use Krebs cycle IL-10 regulation of inflammasome in macrophages is metabolic-dependent and affects DSS-induced colitis development Sema 6D-deficient mice show impaired macrophage polarisation and exacerbated DSS-induced colitis	[118,119,120]

## Data Availability

Not applicable.

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
