# Peer review of "New Perspectives in the Study of Intestinal Inflammation: Focus on the Resolution of Inflammation"

_ijms, 2021, doi:10.3390/ijms22052605_

Round 1

Reviewer 1 Report

Major points

  1. Current literature has revealed the diverse roles of MMPs in IBD pathogenesis. Their regulatory functions in processing chemokine gradient may be the consequence of their effects on epithelial barrier function and immune responses. Therefore, the part of MMPs in section 4.3 should be revised.

Minor points

  1. In Section 2.2, it would be great to include references describing three general aspects of resolution in P3, line 98-114
  2. P3, line 117, the full name of SPM is “specialized” pro-resolving lipid mediators.
  3. P3, line 127, the full name of DPA should be described.
  4. Reference 19 does not describe the regulation of neutrophil adhesion to the endothelium.
  5. P3, line 132, it would be clearer to name other acute models of inflammation that ANXA1 is involved.
  6. Line 150-151, what are the cell sources of FasL and TRAIL that induce the apoptosis of neutrophils in the intestine?
  7. P4, line 154-155, the full name of TIM4 and BAI1 should be described.
  8. P5, line 217, is reference 49 correct? It should be Proc Natl Acad Sci U S A. 2004 Nov 2; 101(44): 15736–15741.
  9. P5, line 223, it is not necessary to abbreviate wild type as wt, especially no more wt is mentioned in this manuscript.
  10. 4.4 and 4.5 are cellular factors that are different from the molecular factors describing in 4.1~4.3. It would be clearer if 4.4 and 4.5 are described in another section, eg. 5.1 and 5.2.
  11. P16, line 555, the reference number is missing.
  12. The authors should check again whether the full names of abbreviations listed in this manuscript are described. 

Author Response

Response to reviewer 1:

Major points

  1. Current literature has revealed the diverse roles of MMPs in IBD pathogenesis. Their regulatory functions in processing chemokine gradient may be the consequence of their effects on epithelial barrier function and immune responses. Therefore, the part of MMPs in section 4.3 should be revised.

First, we would like to thank the reviewer for his/her comments and suggestions, which helped us to improve our manuscript.

Regarding this point, we focused on the specific roles played by different MMPs in the regulation of resolution of inflammation, which based on the evidence we have, could be by mediating by the regulation of chemokine function. It is true that MMPs could control IBD through different mechanisms, as the reviewer mention (regulation of the intestinal epithelial barrier or immune responses). However, at present, there is not any study demonstrating that the regulation of chemokine gradient by MMPs is responsible for their effects on the epithelial barrier function.

It does exist more information about the regulation of immune responses and inflammation by MMPs. There are different studies and even review articles showing the participation of these proteases in the control of the induction of inflammation or inflammatory cell production of cytokines. However, this is not a matter of discussion of this review. In lines 363-368, we commented briefly the relevance of MMPs in the regulation of immune responses and we added several interesting cites related to that, but we wanted to focus only on the effects of these proteins during intestinal resolution of inflammation. As far as we are aware, we included in this manuscript all the studies related to MMPs and resolution of inflammation.

Minor points

  1. In Section 2.2, it would be great to include references describing three general aspects of resolution in P3, line 98-114

We added references as suggested, line 106.

  1. P3, line 117, the full name of SPM is “specialized” pro-resolving lipid mediators.

Thanks to the reviewer for his/her observation. We added “specialized”, line 117.

  1. P3, line 127, the full name of DPA should be described.

The full name of DPA was included, line 127

  1. Reference 19 does not describe the regulation of neutrophil adhesion to the endothelium.

We apologize for that, there was a mistake with that reference. We included the correct reference. Reference 20 in this version.

  1. P3, line 132, it would be clearer to name other acute models of inflammation that ANXA1 is involved.

We included more acute models as suggested by the reviewer, line 132.

  1. Line 150-151, what are the cell sources of FasL and TRAIL that induce the apoptosis of neutrophils in the intestine?

Line 150-151 are related to general aspects of resolution of inflammation. For that reason, we did not include specific details of the source of those molecules in the intestine. In general, macrophages or even neutrophils themselves are able to produce both molecules. We added that information in the manuscript, line 152.

  1. P4, line 154-155, the full name of TIM4 and BAI1 should be described.

We added the full name of TIM4 and BAI1 as requested, line 156-157.

  1. P5, line 217, is reference 49 correct? It should be Proc Natl Acad Sci U S A. 2004 Nov 2; 101(44): 15736–15741.

We apologize for that mistake. We corrected that reference. Reference 53 in this version.

  1. P5, line 223, it is not necessary to abbreviate wild type as wt, especially no more wt is mentioned in this manuscript.

We eliminated “wt” from the manuscript, line 224.

  1. 4.4 and 4.5 are cellular factors that are different from the molecular factors describing in 4.1~4.3. It would be clearer if 4.4 and 4.5 are described in another section, eg. 5.1 and 5.2.

We totally understand the reviewer´s point. However, points 4.4 and 4.5 are also factors involved in the resolution of inflammation in IBD. If we change to 5.1 or 5.2 it could seem that these two points are not related to resolution in IBD, which is not the case.

  1. P16, line 555, the reference number is missing.

The reference number was added, and the reference list was updated. Please see the reference list.

  1. The authors should check again whether the full names of abbreviations listed in this manuscript are described. 

We have checked and added several full names across the text. For instance, line 231, 315 and 504.

Reviewer 2 Report

The authors present an overview of research on new perspectives in study on intestinal inflammation. It is an interesting piece of work, but requires few corrections and additions before being published, so that it constitutes a full presentation of the discussed aspect.

  1. Line 35: please provide the quotations at the end of the sentence referring to the indicated research/reports "most of the knowledge." Here, and in some other places quotation should be provided.
  2. Line 76: "by resident cells of the infected or injured tissue." - this point needs to be developed before the authors proceed to neutrophil influx.
  3. A graphical representation of the discussed aspects would greatly enrich the work.
  4. What was the search methodology and which keywords were used? Bearing in mind the reliability of the conducted review, I suggest incorporating a Materials and Methods section and leading it in the form of a systematic review in order to fully characterize the inclusion and exclusion criteria for manuscripts, indicate the date range of the search, etc. I recommend that the authors consult the PRISMA guidelines to report the work as a systematic review.
  5. Considering the aim of the study, it seems reasonable to extend the review to include additional aspects regulating the course of intestinal inflammation. I think that we should not ignore, for example, the role of alkaline phosphatase or intensively developing research on the influence of exercise and myokines/adipomiokines.
  6. Abbreviations should be defined at the first mention in both abstract and the main text and also below the tables.

Author Response

Response to reviewer 2:

The authors present an overview of research on new perspectives in study on intestinal inflammation. It is an interesting piece of work, but requires few corrections and additions before being published, so that it constitutes a full presentation of the discussed aspect.

  1. Line 35: please provide the quotations at the end of the sentence referring to the indicated research/reports "most of the knowledge." Here, and in some other places quotation should be provided.

We thank the reviewer for his/her comment. However, we would like to comment that we used “most of the knowledge” with the intention of highlighting that a big part of the studies on the resolution of inflammation are focused on the role of SPMs. This is just an introductory comment, we think that adding the quotations in the introduction would be less appealing for the reader. Given the fact that all this information is explained in the indicated sections and tables.

We could not find a similar phrase in other parts of the text.

  1. Line 76: "by resident cells of the infected or injured tissue." - this point needs to be developed before the authors proceed to neutrophil influx.

We added some information in that point as requested by the reviewer, line 76. However, we would like to highlight that this paragraph is a brief and introductory description of the inflammation process. The main topic of the review is the resolution of the inflammation process. We did not want to describe in detail processes that are not relevant to the topic of the review.

  1. A graphical representation of the discussed aspects would greatly enrich the work.

We would like to thank the reviewer for his/her suggestion. As requested, we included a graphical representation. Please see figure 1.

  1. What was the search methodology and which keywords were used? Bearing in mind the reliability of the conducted review, I suggest incorporating a Materials and Methods section and leading it in the form of a systematic review in order to fully characterize the inclusion and exclusion criteria for manuscripts, indicate the date range of the search, etc. I recommend that the authors consult the PRISMA guidelines to report the work as a systematic review.

We understand and appreciate the reviewer´s suggestions. However, we do not agree that adding all this information would increase the quality or significantly improve the content of the manuscript. Our review article cannot be considered as a systematic review or meta-analysis. For that reason, we believe that we should follow the rules and format for standard reviews.

  1. Considering the aim of the study, it seems reasonable to extend the review to include additional aspects regulating the course of intestinal inflammation. I think that we should not ignore, for example, the role of alkaline phosphatase or intensively developing research on the influence of exercise and myokines/adipomiokines.

Again, we would like to thank the reviewer for his/her comment. Our manuscript contains information from factors that are known to be involved in the resolution of the inflammation process. We put a special emphasis on those mediators that according to the existing literature affect intestinal resolution of inflammation. However, there is very limited data (practically none) about the participation of alkaline phosphatase or myokines in the resolution of inflammation and there is any information relating these two factors (or adipomyokines) with the resolution process at the intestinal level.

  1. Abbreviations should be defined at the first mention in both abstract and the main text and also below the tables.

Abbreviations were defined as suggested, line 287, 414.

Round 2

Reviewer 1 Report

No further comments.

Reviewer 2 Report

I have no additional comments.